# Efficient end-to-end learning for cell segmentation with machine generated weak annotations

Prem Shrestha [1], Nicholas Kuang [1] & Ji Yu[1✉]

Automated cell segmentation from optical microscopy images is usually the first step in the pipeline of single-cell analysis. Recently, deep-learning based algorithms have shown superior performances for the cell segmentation tasks. However, a disadvantage of deep-learning is the requirement for a large amount of fully annotated training data, which is costly to generate. Weakly-supervised and self-supervised learning is an active research area, but often the model accuracy is inversely correlated with the amount of annotation information provided. Here we focus on a specific subtype of weak annotations, which can be generated programmably from experimental data, thus allowing for more annotation information content without sacrificing the annotation speed. We designed a new model architecture for end-to-end training using such incomplete annotations. We have benchmarked our method on a variety of publicly available datasets, covering both fluorescence and bright-field imaging modality. We additionally tested our method on a microscopy dataset generated by us, using machine-generated annotations. The results demonstrated that our models trained under weak supervision can achieve segmentation accuracy competitive to, and in some cases, surpassing, state-of-the-art models trained under full supervision. Therefore, our method can be a practical alternative to the established full-supervision methods.

[1]UConn Health, 263 Farmington Ave, Farmington, CT, USA. ✉email: jyu@uchc.edu

In the recent literature, convolutional neural-network (CNN) based deep-learning models[1,2] have demonstrated unparalleled performance in various machine vision tasks, and are increasingly being adopted by biomedical researchers as the method of choice for single-cell segmentation from microscopy images[3]. Currently, there are two general approaches to building deep-learning models for single-cell segmentation. The first is to treat the problem as a pixel-level classification/regression task. In the simplest case, the model takes a microscopy image as input and assigns each pixel a unique class label (e.g., cell vs background) and thus produces a segmentation mask for the input image. Because the image can have more than one cell, a post-processing step, typically based on a simple morphological operator, such as watershed[4], is employed to break the segmentation mask into instances of individual cells[5,6]. Unfortunately, this only works well for cells at low density or for nuclei segmentations[7], but cannot handle dense cell populations. Predicting cell border pixels as a distinct class[8,9] serves as a simple work-around, albeit with the caveat that it is often ambiguous to which cell one should assign the border pixels. A more popular solution for a true single-cell segmentation algorithm is to train the model to predict a distance metric[10–16] for each pixel, e.g., the Euclidean distance to the nearest background pixel. The resulting "distance map" can then be processed to produce segmentations for individual cells. The various models currently in the literature differ in the exact distance metrics they choose to compute. Nevertheless, a common characteristic of this general approach is that the segmentation proceeds as a two-step process. In the first step, a deep-learning model is used to convert a microscopy image to an intermediate pseudo-image that is more algorithmically manageable. Then at a second step, a hand-crafted post-processing algorithm is used to convert the pseudo-image into instance segmentations of single cells.

The second general approach to cell segmentation has its roots in the object-detection literature from the computer science field. In this case, a deep-learning model tries to perform the joint tasks of object detection, object segmentation and object classification concurrently in an "end-to-end" fashion. Instead of a pseudo-image, the model was trained to directly output a list of segmentation masks, each representing a single object (i.e., a cell) in the input image. Therefore, there are no post-processing steps. In addition, the Individual object masks produced by these models are allowed to have spatial overlaps, which in principle allow them to better handle more complicated segmentation problems where cells crawl on top of each other. Many general-purpose instance segmentation models of this type have already been developed, e.g. MaskRCNN[17], CenterMask[18] and YOLO[19]. Although these models were not designed specifically with biological applications in mind, several studies have already successfully shown that they can be apply to microscopy image data to solve cell segmentation problems[20,21].

Regardless of the specific approaches, a well-known disadvantage of the deep-learning methods is that they are very data-hungry and require a large amount of annotated training data. For example, a recent study showed that the accuracy of the cell segmentation models did not reach saturation even after training with >1.6 million cell instances[20]. Training data for single cell segmentation are particularly costly to generate because the annotations must be produced at the instance level, i.e., the exact boundaries of each cell need to be manually determined, unlike many other machine vision tasks such as image classification, where the annotations were produced at the image level. This drawback further raises concerns over the scalability of the deep-learning method for three-dimensional (3D) microscopy, for which manual annotations are even more expensive to produce.

To alleviate the burden of image annotation, a lot of effort has been put into the studies of weakly-supervised learning[22] and self-supervised learning[23] from images, particularly within the general instance segmentation literature. For example, it has been shown that a MaskRCNN-type model can be successfully trained with only bounding-box annotations, achieving more than 90% of the performance relative to full-supervision (i.e., with instance masks)[22]. Combining bounding-box supervision with additional randomly-sampled point supervision resulted in even better performances[24,25] and this latter strategy has already been tested on single-cell segmentation tasks[26]. However, per-instance bounding-box annotation is still time-consuming. Location-of-interest (LOI)[27], which labels the approximate location of an object, is easier to produce but also is a weaker form of supervision than either the bounding box or the point supervision. A method to train instance segmentation models using LOI supervision has been proposed, but the model performance is lower[27]. An attractive alternative approach is to rely on image-level annotations, instead of instance-level annotations[28–31]. For example, class-activation maps[32] can be utilized as a stand-in for segmentation masks[28–30] because high gradients tend to localize to regions of importance for the modeling objective. This allows the model to be trained on simpler auxiliary tasks, such as image classification. These approaches have achieved good results for semantic segmentation tasks, but applying them to dense instance segmentation, i.e. for single-cell segmentation, remains unproven. Finally, advances have also been made in self-supervised learning[33–37], which focuses on learning a useful representation of the image features without focusing on a specific task or using any labels – the learned representations can then be used for more specific downstream tasks, including image segmentation. Another revenue of research is to incorporate specific prior knowledge into models built for specific microscopy modalities. For example, Hou et al.[38] were able to model H&E images by combining a sparse autoencoder to represent nuclei blobs and a light-weight CNN to represent background, and extracted useful feature representations based on unsupervised training.

Faster annotation and higher model accuracy, however, are often two conflicting goals. The model accuracy generally improves with the more detailed annotations, which requires longer time to generate, and vice versa. Therefore, in this paper, we will focus on cell segmentation models trained with weak annotations that potentially can be produced programmably or semi-programmably from experimental data. We believe this strategy is the best compromise between the goals of high model accuracy and fast annotation (Fig. 1a). Specifically, we will focus on two types of annotations: (1) Image-level segmentations (Fig. 1b), which separates the cellular region from the background region. Such an annotation can be produced by acquiring a fluorescence image of the cellular sample and thresholding the resulting image. (2) Location-of-interests (LOIs), which are the rough locations for each cell present in the image (Fig. 1b). This annotation can be produced by acquiring nucleus images using widely employed labels, such as diamidino-2-phenylindole (DAPI). One way to annotate LOIs is to apply an existing nuclei segmentation model[7], although it should be noted that LOIs is a location label and can also be obtained without fully segment the nuclei, by via simpler blob detection algorithms. By focusing on these two specific annotations, a researcher can potentially generate a large dataset quickly, and still offer annotations with rich informational content for training an accurate segmentation model.

## Results

**Model design**. Figure 2a shows our model architecture for the single-cell segmentation task. The overall model contains three sub-modules, all of which are based on fully convolutional neural

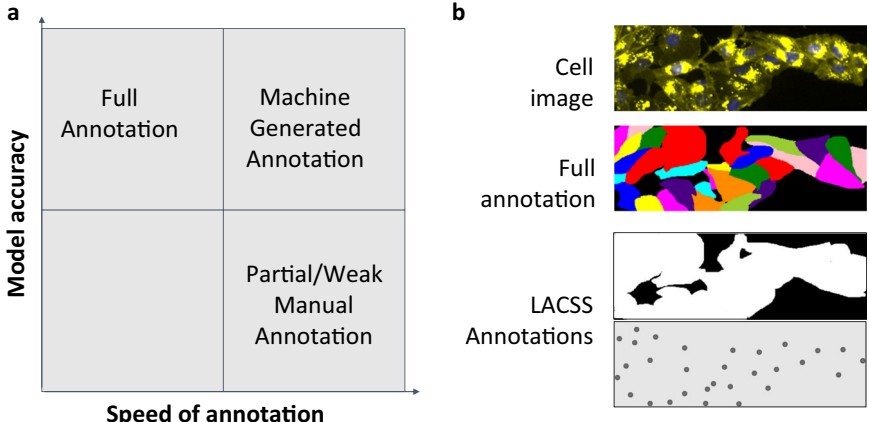

**Fig. 1 Comparison of various training data annotations for single-cell segmentation models. a** Diagram illustrating the conflicting goals of faster annotation and higher model accuracy, and how experimentally acquired annotations offer a better compromise. **b** Examples of annotations. The top image shows a representative microscopy image of cells. The second row shows the corresponding full annotation of single-cell segmentations, which is expensive to generate. The bottom two rows showed two types of incomplete annotations that can potentially be generate programmably: (1) image-level segmentation, which labels the pixels of all cells in the image, and (2) location-of-interests annotation, which a subtype of the point annotation that denotes rough locations of individual cells.

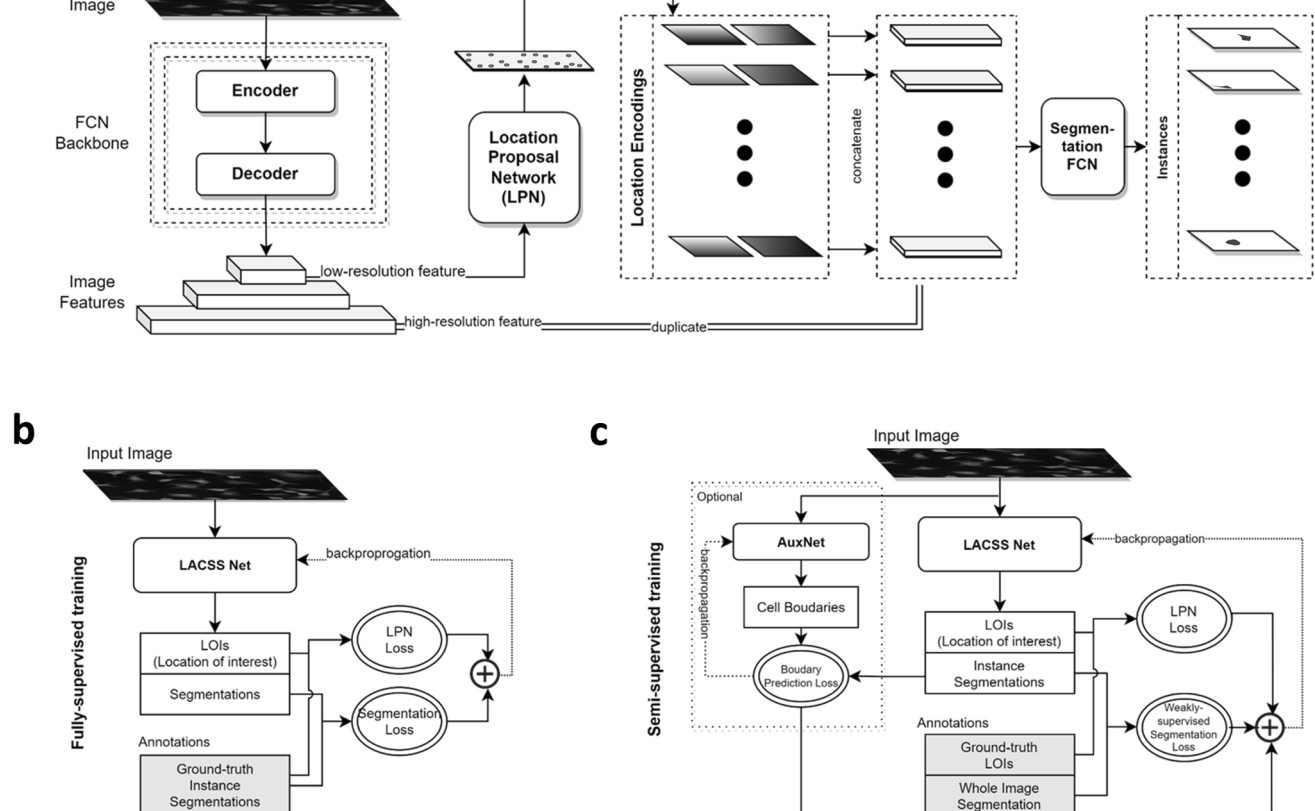

**Fig. 2 LACSS model. a** Model architecture showing all three components of the model: backbone, location proposal network (LPN) and segmentation FCN. **b** Schematic diagram of LACSS model training under full-supervision. **c** Schematic diagram of LACSS model training under semi-supervision.

network (FCN) design. The backbone employs a standard encoder/decoder structure, and outputs feature representations extracted from the input image at multiple resolutions. The second component, the location proposal network (LPN), is analogous to the region proposal network (RPN) of the popular FastRCNN model for object detection[39]. Its task is to predict a set of LOIs from the feature outputs of the backbone. Unlike RPN, which computes regression for the object bounding-box (location and size), the LPN does not predict the object size because the information is unavailable in the annotations. The last component, the segmentation FCN, is responsible for output the single-cell segmentations, taking the high-resolution image features as the input. To allow the network to focus on a specific cell, the image features were first concatenated with small tensors

encoding the predicted LOIs, one for each cell (Supplementary Note 1. Fig. S1). This way, the segmentation FCN receives different inputs for different cells, and thus can produce variable segmentations even though the image features are constant. To increase the algorithm efficiency, the segmentation FCN only performs computation within a small area around the LOI, as it can reasonably assume that pixels far away from the LOI are not part of the segmentation (Supplementary Note 1 and 2). Combined, the three modules learn the segmentation task in an end-to-end manner. We name our network architecture location assisted cell segmentation system (LACSS).

Even though the model is designed with incomplete annotation in mind, it is also able to learn under full supervision (Fig. 2b, Supplementary Note 2). Under such a configuration, the model loss is the sum of two components. The first is the LPN loss, which measures the differences between the predicted LOIs and the ground truth LOIs. The latter can be computed from the individual cells' ground truth segmentations. The second is the segmentation loss, which is simply the cross-entropy loss between the predicted cell segmentations and the ground truth cell segmentations. On the other hand, if the model is trained under the configuration of the weak supervision (Fig. 2c), the LPN loss can still be calculated because the LOIs are part of the image annotation. The segmentation loss, on the contrary, is no longer available. Therefore, we used instead a weakly-supervised segmentation loss function (Supplementary Note 2), which computes (i) the consistency between single-cell segmentation with the image-level segmentation, and (ii) the consistency between individual cell segmentations. The latter part aims to minimize overlaps between individual cell segmentations. Detailed mathematical formulations of all loss functions are provided in Supplementary Note 2.

**Cell image library dataset.** As a proof of principle, we first performed a quick segmentation test on the publicly available Cell Image Library[40] dataset (Fig. 3). The dataset contains high quality two-channel fluorescence images of neuroblastoma cells, fully annotated with single-cell segmentation masks. We first converted the original cell segmentations to image-level segmentations by combining all cell masks and used the center-of-mass of the individual cells as the LOIs. We then trained weakly supervised LACSS models using the generated annotations.

To benchmark the segmentation accuracy, we computed average precision (AP) on the validation set, as well as the aggregated Jaccard index (AJI). The APs were computed under a series of intersection-over-unions (IOUs) criteria ranging from 50% to 95% (Fig. 3). The model trainings were run for multiple times ($n = 5$). We found the best $AP_{50}$ score surpassed 0.91 in all runs, with the best model reached $AP_{50} = 0.933$ and $AJI = 0.751$. The AP values at higher IOU criteria and example segmentation predictions are shown in Fig. 3.

Both the quantitative benchmarks and visual inspection of the segmentation results (Fig. 3a) suggested that our model performs well on this dataset. To our surprise, our $AP_{50}$ seem to be better than previous reports based on segmentation models training with full supervision[15]. Using the exact same dataset, Stringer et al. reported APs ($IOU = 50\%$) from three different segmentation models (cellpose[15], MaskRCNN[17]s and stardist[11]), all of which are lower than ours by more than ten percent points. We also trained LACSS with full supervision and obtained similar accuracy (best $AP_{50} = 0.937$) as the semi-supervised training. However, we did observe tendency to overfit under full-supervision probably due to the small size of the dataset, which may have contributed to the poorer results reported previously.

**LIVECell dataset.** Encouraged by the positive outcome on the cell image library data, we then moved on to test our model performances on a much larger dataset, LIVECell[20]. This is a recently published cell segmentation dataset containing >1.6 million fully segmented cells from eight different cell lines. The imaging modality is bright field, which has lower contrast than fluorescence for the purpose of cell detection. The cell lines chosen covered a diverse set of morphologies. Cell-cell contacts and cell-cell occlusions are of frequent occurrences. Therefore, this is a more difficult study case for cell segmentation but is also more representative of real-life problems in biomedical research. In addition, baseline benchmarks were already provided based on two state-of-the-art models, MaskRCNN[17] and CenterMask[18], trained on full-supervision with human annotated instance segmentations.

Similar to the Cell Image Library case, we first converted the original instance-level annotations to image-level segmentations and LOIs. We then trained LACSS models with the converted weak annotations. We also train LACSS with the original full-supervision in order to understand the performance gaps between the two configurations. We found that indeed LACSS trained under weak-supervision underperform by a quite large margin (Fig. 4). Among the eight cell lines, the SkBr3 line exhibited the smallest performance gap ($AP_{50} = 0.857$ / 0.948 for weakly-supervised/fully-supervised models), and the SHSY5Y line showed the largest gap ($AP_{50} = 0.128$ / 0.520).

Manual examination of the segmentation results indicated that the main weakness of the semi-supervised models is with the determination of the exact cell-cell boundary (Supplementary Fig. S2). Indeed, we noticed that the weakly-supervised model can correctly infer the cell locations as well as the rough sizes and shapes of the cell but has trouble discerning the detailed shape of the cell border. To correct this issue, we introduced an auxiliary convolutional network (auxnet) during the training specifically to predict the cell boundaries (Fig. 2c). Of course, the auxnet cannot be directly trained against ground truth, as the cell boundaries are not part of the annotation. Therefore, we train auxnet against the segmentation output from the main LACSS network (Fig. 2c), and the auxnet predictions in turn help to train the LACSS model weights by constraining LACSS outputs. Auxnet is only used during training and is not part of the computation during inference.

Importantly, we designed the auxnet to be of low field-of-view (FOV). The rationale is that this is also how humans perform cell segmentations. Typically, we first scan a relatively large area in order to recognize the cell and understand its rough shape, but when it comes to tracing the exact cell boundaries, we will look at a much smaller area in order to find the exact pixel separating two cells. By designing the auxnet to be of low FOV, we force it to focus on a different set of image features than that of LACSS, which has a much larger FOV.

Introducing the auxnet in the training significantly improved the model accuracy (Fig. 4, Supplementary Table S1). For two of the cell lines (BV2 and SkBr3), the differences between the weakly-supervised models and fully-supervised models had almost entirely disappeared as measured by the APs. For the rest of the cell lines, the weakly-supervised models still underperform, but the gaps are much smaller (Fig. 4).

We also compare our results with the previously published baselines (Table 1). The two baseline models were trained end-to-end with full supervision and employed more complex backbones[41] (ResNet200) than ours (ResNet50). Nevertheless, we found that our results were competitive to these previous models. For BV2 and SkBr3 lines, our model had $AP_{50}$ scores slightly higher than previous baselines. When using more stringent IOU criteria (e.g., >=75%) our results are less

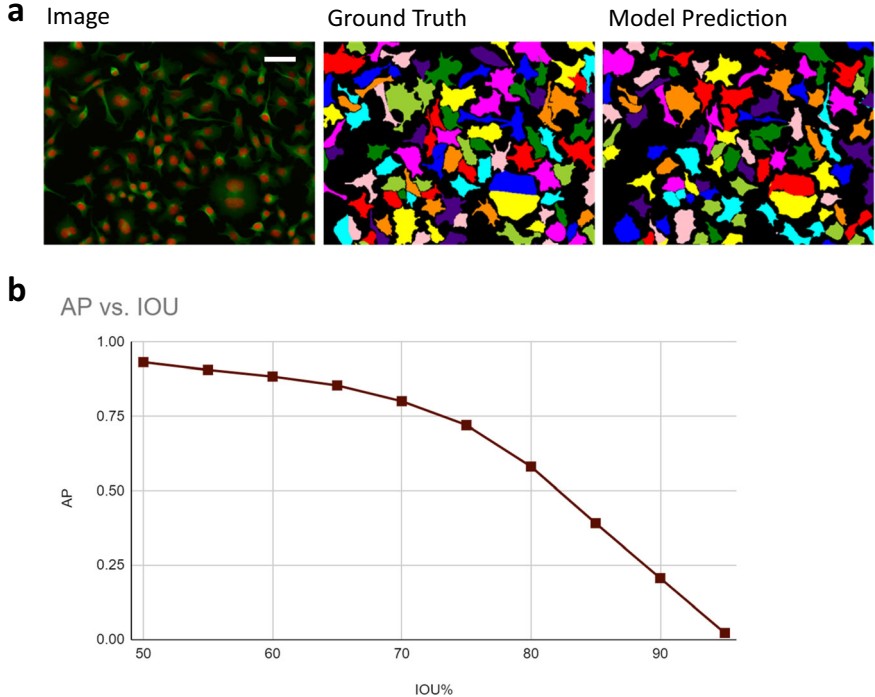

**Fig. 3 Segmentation of cell image library dataset with a LACSS model trained with incomplete annotations. a** A representative example of segmentation results showing the input image (left), the ground-truth segmentation (middle) and model prediction (right). The pseudo colors were generated via skimage's label2rgb function for visual clarity and carried no extra meaning. **b** Model performances quantified by average precisions and false negative rates at various IOU thresholds. Scale bar represents 50 micrometers.

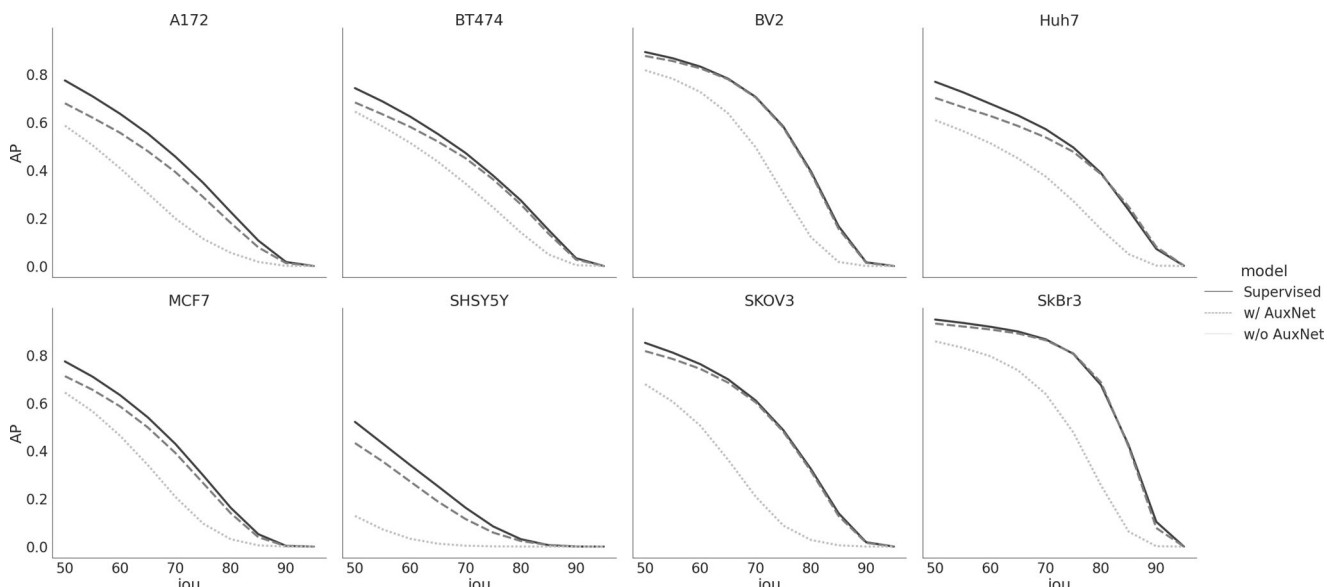

**Fig. 4 LACSS model accuracy trained with LIVECell dataset.** Models were trained either under full-supervision using the original single-cell annotation (supervised, solid line), under semi-supervision using synthetic annotation but without auxnet (dotted line), or under semi-supervision with boundary-defining auxnet (dash line). AP values were computed for eight cell lines at increasing IOU thresholds from 50% to 95%. The results showed significant improvements in model accuracy by introducing auxnet in the training for models trained with semi-supervision.

competitive (Table 1). Similar trend can be seen in comparison of AJI values (Table 1), indicating LACSS has difficulties achieving pixel-perfect segmentations.

Our results showed the highest performance gap in the SHSY5Y cell line (Table 1). Interestingly, for this cell line, the LACSS model performs poorly even when trained under full supervision ($AP_{50} = 0.520$), suggesting that the weakness stems from the model architecture and not from the training strategy.

The exact reason why our model architecture performed poorer on SHSY5Y is not fully understood yet and will be examined in the future.

Figure 5 showed examples of segmentation results on the eight cell lines. The model generally was able to trace the boundary between cell-cell contacts, despite the fact the training data contains no such information. The model also recaptured certain annotation choices to a degree. For example, for BV2 cells, the

**Table 1 Comparison of LACSS model and the baselines on the LIVECell dataset.**

| Cell lines | MaskRCNN (fully-supervised) | | | | CenerMask (fully-supervised) | | | | LACSS (semi-supervised) | | | |
|---|---|---|---|---|---|---|---|---|---|---|---|---|
| | AP50 | AP75 | mAP | AJI | AP50 | AP75 | mAP | AJI | AP50 | AP75 | mAP | AJI |
| A172 | 0.74 | 0.36 | 0.38 | 0.59 | 0.72 | 0.30 | 0.35 | 0.59 | 0.68 | 0.29 | 0.33 | 0.54 |
| BT474 | 0.76 | 0.44 | 0.43 | 0.56 | 0.78 | 0.43 | 0.43 | 0.52 | 0.68 | 0.36 | 0.37 | 0.53 |
| BV2 | 0.85 | 0.60 | 0.53 | 0.68 | 0.85 | 0.45 | 0.46 | 0.67 | **0.88** | 0.58 | 0.52 | 0.63 |
| Huh7 | 0.80 | 0.58 | 0.52 | 0.64 | 0.78 | 0.50 | 0.46 | 0.64 | 0.70 | 0.48 | 0.43 | 0.60 |
| MCF7 | 0.74 | 0.36 | 0.38 | 0.53 | 0.76 | 0.33 | 0.37 | 0.57 | 0.71 | 0.27 | 0.33 | 0.53 |
| SHSY5Y | 0.61 | 0.16 | 0.25 | 0.49 | 0.61 | 0.14 | 0.24 | 0.50 | 0.43 | 0.06 | 0.14 | 0.40 |
| SKOV3 | 0.87 | 0.60 | 0.54 | 0.73 | 0.86 | 0.53 | 0.49 | 0.71 | 0.82 | 0.48 | 0.46 | 0.66 |
| SkBr3 | 0.91 | 0.80 | 0.65 | 0.81 | 0.90 | 0.80 | 0.66 | 0.79 | **0.93** | **0.81** | 0.65 | 0.75 |

All baseline models were trained with the full annotation, whereas LACSS models were trained using the incomplete annotation. Bold font highlights categories for which the LACSS outperformed the baselines.

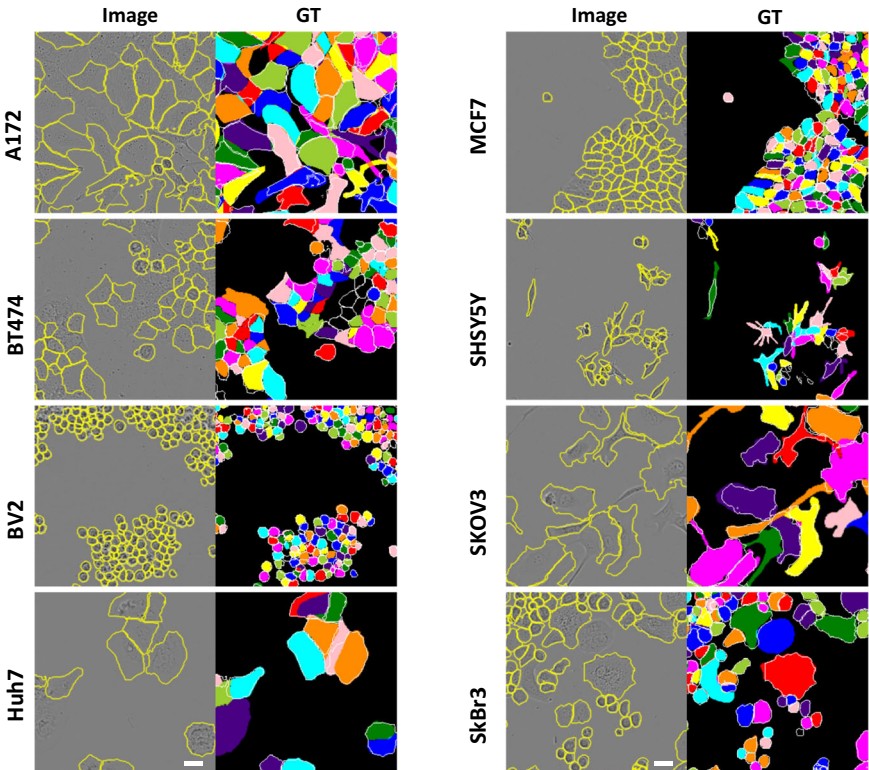

**Fig. 5 Segmentation examples for LIVECell dataset using LACSS models trained with the semi-supervision configuration.** Examples for all eight cell lines were shown. The input phase contrast images (left) were overlaid with model predictions drawn as yellow contours. The ground truth segmentations were shown on the right, for which the model predictions were also shown as white contours for comparison. Scale bars represent 20 micrometers.

annotator chose not to label dying cells with an abnormal appearance, and the model similarity avoided their detections. On the contrary, for BT474 cells, the annotator frequently avoided labeling cells at the center of the colonies, probably due to poor contrast. However, this preference was not learned by the model.

**TissueNet dataset**. To further examine the generalizability of the LACSS approach, we tested our model on the TissueNet dataset, which is a fully-annotated dataset focusing on tissue imaging data[16]. Like in the case for LIVECell, we ignored the original single-cell masks and generated new annotations of LOIs and image-level segmentations to train a semi-supervised LACSS model (Fig. 6). The best model achieved $AP_{50} = 0.78$ and $AJI = 0.66$ (Table 2 and Supplementary Fig. S3). In addition, we also evaluated custom defined precision, recall and F1 score (harmonic mean of precision and recall) according to the original

publication[16], so that we can compare our results with previous published baseline (Table 2). We found that our model achieved about 95% of the performance in comparison to the fully-supervised model measured by the F1 score (0.78 vs 0.82). More detailed examinations showed that the main deficiency of our model is with lower recall, while the model accuracy is similar to the fully-supervised model.

**LACSS with machine generated annotations**. Finally, we turn to test a training pipeline, in which the annotation is generated semi-programmably (Supplementary Fig. S4). To do that, we acquired a dataset on cultured A431 cells using fluorescence microscopy. The cell line is chosen for its tendency to grow in large rafts/colonies with closely packed cells, which means that there will be significant loss of information going from the instance segmentation annotation to the image-level segmentation. This allows for a

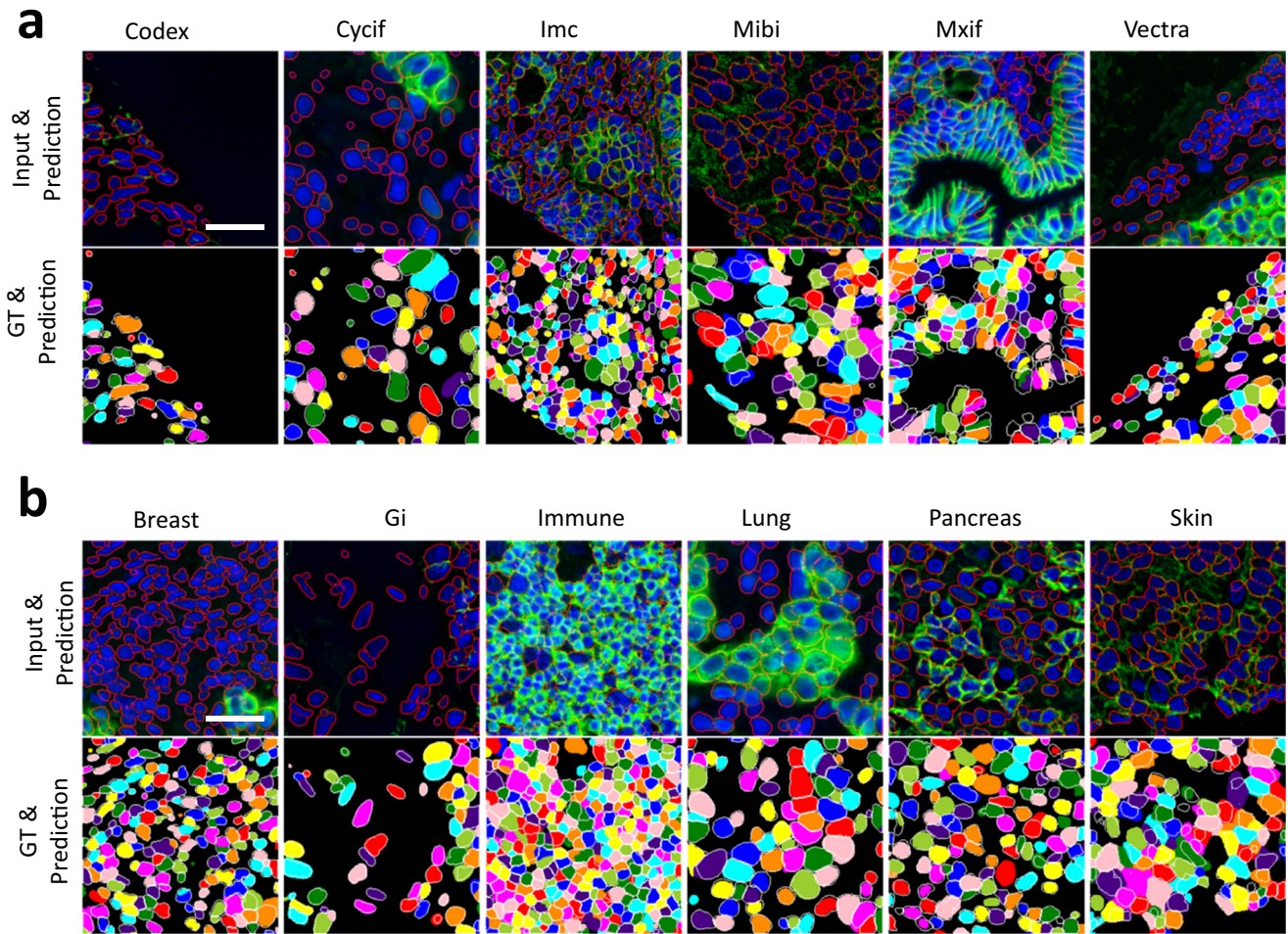

**Fig. 6 Segmentation examples for TissueNet dataset using LACSS models trained with the semi-supervision configuration. a** Example segmentations of representative images from the test split, one for each of the six different imaging platforms. **b** Example segmentations of images from the six different tissue sources. The top rows show the two-channel input images (blue: nucleus, green: membrane) with the segmentation results plotted as contour overlay in red. The bottom rows show the ground-truth segmentations by human annotators, with the model prediction plotted as the contour overlay in white. Scale bars represent 50 micrometers.

**Table 2 Benchmarks of the semi-supervised LACSS model on the TissueNet dataset.**

|  | F1 | Precision | Recall | AJI | AP50 | AP75 | mAP |
|---|---|---|---|---|---|---|---|
| Mesmer | 0.82 | 0.83 | 0.81 | 0.71 | - | - | - |
| LACSS | 0.78 | 0.84 | 0.73 | 0.66 | 0.78 | 0.42 | 042 |

The results from the Mesmer model (fully-supervised) are shown for comparison.

more stringent test of our method. We acquired 500 images (512 × 512 pixels) of the sample labeled with anti-PY100 (Fig. 7), which served as the training set. We also acquired DAPI images of the same cells to help producing the LOI annotations. Furthermore, using the same protocol but in a separate experiment, we acquired 25 more fluorescence images, which we manually segmented to serve as the validation set.

We computed LOIs for the training set from the DAPI images using a simple blob detection algorithm based on the difference-of-Gaussian filter. Much more sophisticated algorithms are available[7,42]. However, our strategy minimized the dependencies on external training data. Even with such a simple algorithm, the results were accurate enough: we estimated that both false positive and false negative rates to be around 1-2% based on the

visual inspections. We also opted to not manually correct these mistakes, in order to evaluate the model performance in a more streamlined pipeline. To generate image-level segmentations, we used an existing graph-cut software (see methods for details). Figure 7a shows two examples of such machine generated annotations. It took a total of several hours to generate and validate the annotations for the whole training set.

In Fig. 7b we showed representative examples of the LACSS segmentation results. Visual examining confirms that the trained model produced segmentations qualitatively consistent with manual segmentation. Evaluation of the model showed that it achieved $AP_{50} = 0.84$ (Table 3, Fig. S5).

Our results on the A431 dataset confirm that the exact nature of the LOIs can be flexible. In previous tests, we used the locations of the center-of-mass of cell segmentations, because the dataset was already pre-segmented. Here the nucleus locations were typically off-center to the cell and randomly localized. Neither are the LOIs center of the nucleus, as the results depend on the exact intensity distributions of the DAPI signal. However, these features did not seem to prevent training of the segmentation model.

## Discussion

Annotation burden is a well-known bottleneck of deep learning methods. The issue is particularly challenging for single-cell

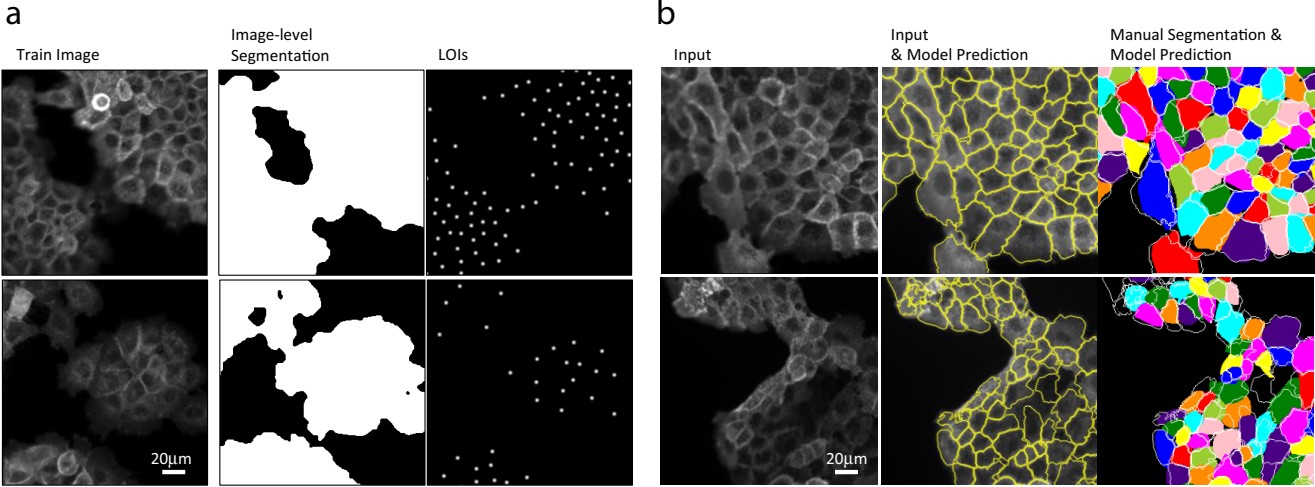

**Fig. 7 LACSS model trained with semi-programmably generated annotations. a** Examples of training data and accompanying annotations generated semi-programmably. **b** Examples of model inferences. The left panel shows the input fluorescence image. The middle panels show the comparison of input image and the model predictions (yellow contours). The right panel shows the comparison of model prediction (white contour) with manual segmentation (color map). The bottom example represented a more challenging case with atypical image contrasts. Scale bars represent 20 micrometers.

**Table 3 Benchmarking LACSS models trained with A431 image data and annotations.**

|       | AP50 | AP75 | mAP  | AJI  |
|-------|------|------|------|------|
| LACSS | 0.84 | 0.28 | 0.39 | 0.65 |

The training dataset includes 500 immunofluorescence images of A431 cells at high density. All annotations were derived from experimental data semi-programmably. See text for more details.

segmentation tasks. One way to solve the problem is to use models that can be trained with weak supervisions, but one needs to strike a balance between the model accuracy and the amounts of annotations. Here, we focus on utilizing annotations that can be generated automatically or semi-automatically, leveraging on experimental resources that are commonly available in biological research labs. This allows us to retain as much annotation information as possible, while not being slowed down by manual annotations. We believe this is the best chance to create an efficient model building pipeline, without sacrificing too much about the model accuracy, which is one of the top priorities for most of the biological researchers. We demonstrate here that segmentation models can be built by utilizing two types of weak annotations: the image level segmentation and the location-of-interests, both of which are incomplete descriptions of the cell segmentation but combined provided sufficient information to achieve good model accuracies.

Technologically, the main contributions of this study are twofold. Firstly, we designed a new loss function that allows us to train the network using a weak supervision scheme. Additionally, we also demonstrated that using a low FOV auxnet as a training aid can significantly increase the model performance for the segmentation task. While the general idea of using one CNN to train another CNN is not new (e.g., GAN[43]), to our knowledge this technique has not been used in cell segmentation literature previously. It is interesting to note that in GAN, the discriminator can be viewed as a particularly complicated loss function for training the generator. Analogously, in our LACSS model, the auxnet can be viewed as a complex regularization term to the loss function.

While our network architecture is derived from MaskRCNN, it is worth pointing out an important distinction: In LACSS the segmentation network operates on a linear combination of image features with positional encodings, while MaskRCNN does not utilize such technique. The positional encodings are needed in order to break the translational invariance of the CNN architecture, without which the model will have no concept of different cells. MaskRCNN does not need such a component because the segmentation is performed only within the bounding box of each object. LACSS, on the other hand, has no access to the bounding box information and relies on a different strategy. We demonstrated here that a light-weight linear encoding scheme is sufficient to achieve this goal.

We have tested LACSS on four different datasets with varying imaging modalities and sample characteristics. The overall results confirmed that the approach is generalizable and not limited to specific types of data. In several cases, the accuracy of LACSS is similar or close to fully-supervised models, but large performance gaps still exist for certain samples, e.g., SHSY5Y cells of LIVECell dataset. We also noticed that if a fully-supervised model can perform well on certain data (e.g., SKBr3 cell), then LACSS can often perform equally well. On the other hand, if a fully-supervised model already struggles with the data (e.g., SHSY5Y cell), then LACSS seems to perform even worse.

Comparing to fully supervised models, the semi-supervised LACSS has relative weakness at higher IOU criteria (i.e. IOU >= 0.75), which is probably not too surprising. Greenwald et al.[16] had asked different human annotators to re-segment the TissueNet data, and compared the results against the "ground truth". They found that new human annotators typically achieve a score (F1~0.65) much lower than the fully-supervised CNN models (F1 > 0.8). This suggests that part of the model performance should be attributed to its ability to reproduce the annotator biases/preferences the model was trained on. For weakly supervised models, fully recapitulating annotator biases might be quite difficult because they don't have full access to all the annotations. We suspect this factor partly explains the performance gap between LACSS and other fully-supervised models.

Our goal is to build a streamlined annotation-training pipeline that requires little to no manual input from humans. Our demonstration on the A431 fluorescence dataset indicated that this is indeed feasible. However, the requirement of image-level segmentation annotation remains to be somewhat of a bottleneck. Image-level mask is relatively easy to obtain when fluorescence

data are available, but there are cases when this condition is not easily met, e.g., histological samples. It is much preferable if a model can be trained using only the LOI annotations. It should be noted that the LOI annotation is very different from the so-called 'point' annotation that has already successfully been used in the literature[24,26]. The "point" annotations are *randomly* sampled points from a segmentation mask. The randomness is critical in order to provide sufficient shape information regarding the mask. In addition, typically both positive samples and negative samples are needed for point annotation to be effective. The LOI annotation, on the contrary, has no such constraints, and is thus a much weaker form of supervision. Indeed, training segmentation models using LOI remains to be a challenge and will be the focus of our future efforts.

In conclusion, we provided a practical alternative for efficiently building single-cell segmentation deep-learning models, which would be of interest to biological researchers in need of performing single-cell analysis from microscopy images.

## Methods

**Image acquisition**. For this study we experimentally acquired a small microscopy dataset on the human squamous-cell carcinoma line A431 (ATCC CRL1555). Cells were seeded on glass substrate and grown to ~80% confluence in standard growth media before fixation for imaging. Cells were labeled with the standard immuno-fluorescence protocol. They were washed with a blocking buffer (2% BSA in PBS, 1% TX100), incubated with Alexa647-labeled anti-PY100 (1:500 in blocking buffer containing DAPI) for 2 h while rocking at 4 °C. Images were acquired on an automated inverted fluorescence microscope (Olympus IX) with an 20x objective. Images were captured on a camera with a $512 \times 512$ sensor. The per pixel dimension is 0.8 micrometer under this configuration.

**Image annotation**. Only the lab acquired A431 cell data were annotated by us. All other dataset used in this study was previously segmented manually. To annotate the A431 training set, we use the existing GraphCut function of the FIJI software[44] to produce image-level segmentation. All images were segmented at five different intensity thresholds and a human annotator later manually examined the output to pick one out of five for each input image. To produce LOIs, we first apply a difference-of-Gaussian filter to the DAPI images, with the σ values of 4.2 and 5 pixels. We then searched for all local maxima with four-connectivity and used their locations as LOIs. To annotate images in the validation set, we used the online annotation software described by Stringer et al.[15] to generate single-cell segmentations for all cells in the image. The software takes polygon input indicated by mouse clicks and converts the inputs into single-cell segmentations.

**Datasets**. In addition to the lab-generated A431 dataset, we used three published cell segmentation datasets: Cell Image Library[40], LIVECell[20], and TissueNet[16]. Cell Image Library dataset contains 100 dual channel fluorescence images of neuro-blastoma cells. We split the images into training ($n = 89$) and validation ($n = 11$) sets, following Stringer et al.[15]. The LIVECell dataset consists of fully segmented bright field images of eight different cell lines. The original authors had pre-split the dataset into training (3253 images), validation (570 images), and testing (1564 images) sets. TissueNet data consists of dual-channel fluorescence images of histological tissue samples from six different tissue types, acquired using six different imaging platforms. Additionally, samples were sourced from six different tissue types, representing a wide variety of cell morphologies and organizations. The dataset was also pre-split into training (2601 images of $512 \times 512$ pixels), validation (3104 images of $256 \times 256$ pixels) and testing (1249 images of $256 \times 256$ pixels) sets. For both LIVECell and TissueNet, we use the dataset according to the original splits.

**Model training**. Here we provide a general outline of the model training procedure. See Supplementary Note 3 for detailed descriptions of the hyperparameter choices and the training procedures of each model.

Models for the cell image library dataset were all trained with He initialization[45] of model weights. The experiments were repeated five times and the best model was chosen based on the highest $AP_{50}$ score on the validation set. Models for LIVECell and TissueNet dataset were also trained with the initialization scheme, i.e., without pretraining on external datasets. Best models were chosen based on benchmarks on the validation split of the dataset. The testing set was not used when choosing models. For the lab-acquired A431 dataset, the model was pretrained on the LIVECell dataset, then transferred to training on the A431 data and evaluated on the A431 validation set. All models were trained using a single NVIDIA Tesla A100 GPU. ADAM optimizer was used for all experiments.

**Model benchmarks**. To evaluate model accuracy, we relied primarily on the average precision (AP) metrics, which were widely used in the instance segmentation literature for evaluating ranked list of segmentation predictions:

$$AP = \frac{1}{c}\sum_{k=1}^{n}P(k)T(k) \qquad (1)$$

where $C$ is the total number of ground truth cells, $n$ is the total number of detections, $T(k)$ is an indicator function of whether the $k$-th detection is positive ($T(k) = 1$) or negative ($T(k) = 0$), and $P(k)$ is the precision of the first $k$ detections. Whether a detection is considered positive is determined by the intersection-over-union (IOU) of the detection against the ground truth. We use the notation $AP_{IOU}$ to denote the AP values at a specific IOU threshold, i.e, $AP_{50}$ is the average precision when the positive detection requires a minimal IOU of 50%. We also use the notation mAP to denote the average AP over a series of IOU thresholds. In our study, we chose IOU thresholds from 0.5 to 0.95 (inclusive) in a 0.05 step size. Per convention of instance detection literature, we do not allow multiple detections to match against the same ground truth instance.

A secondary benchmark for segmentation accuracy is aggregated Jaccard index (AJI), as defined in Kumar et al.[46]. AJI is an extension Jaccard index, a pixel level metric, by incorporating object matching to allow it to be used for instance segmentation results.

$$AJI = \frac{\Sigma_i |G_i \bigcap S_i^*|}{\Sigma_i |G_i \bigcup S_i^*| + \Sigma_j |S_j^o|}, \qquad (2)$$

where $G_i$ is a ground truth cell, and $S_i^*$ is the best model prediction of $G_i$, and $S_j^o$ represents predictions that doesn't match any ground truth.

Additionally, for the TissueNet datasets we computed a custom F1 score according to the original publication[16], to facilitate comparison with the previously published baselines. The specific score is a measure for the accuracy of cell detection using an IOU threshold of 0.6 as the criteria for positive detection.

**Statistics and reproducibility**. The study did not employ statistical analysis when comparing LACSS models with published baselines. All reported model benchmarks are based on best models obtained from multiple reruns of the training pipelines. Optimized model weights are available (see below) to allow re-evaluation of the models.

**Reporting summary**. Further information on research design is available in the Nature Portfolio Reporting Summary linked to this article.

## Data availability

The A431 image dataset and the associated annotation is available at Mendeley data repository: https://data.mendeley.com/datasets/89s3ymz5wn/1. Model weights for all LACSS models shown in the paper are available at the LACSS website: https://github.com/jiyuuchc/lacss. Source data for Figs. 3b and 4 can be found in Supplementary Data.

## Code availability

All software source code is available at https://github.com/jiyuuchc/lacss.

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

## Acknowledgements

The research is in part funded by National Institute of Health (R01GM123784). N.K. is support by the UConn HRP program for student research.

## Author contributions

P.S. and N.K. performed experiments. J.Y. designed the research, performed experiments, and wrote the paper.

## Competing interests
The authors declare no competing interests.
