## [Peer Review File · Communications Biology]

Reviewers' comments:

Reviewer #1 (Remarks to the Author):

The manuscript „Efficient End-to-end Learning for Cell Segmentation with Machine Generated Weak Annotations” by Shrestha et al, addresses the current need in cell image segmentation for deep learning algorithms to be trained on human annotator generated ground truths. The authors propose an end-to-end model using machine generated annotations and show that in some microscopy image data series segmentation results are equally or even more accurate as compared to previously published models, such as Cellpose. For some images AP is less than expected.

The main strength of the approach is the author’s notion to keep annotation times short. Further the authors propose to additionally predict cell boundaries using an auxiliary convolutional network which improves the result of weakly-supervised models.

I have two main points of criticism/suggestion:

1. The impact of the work would improve significantly if the value of the model would be tested on tissue images rather than solely cell cultures.

2. Figure 4 shows that at an IoU of 50 – which is rather low – the weakly supervised model plus auxnet (nearly) reaches AP levels of the supervised model. However, an IoU of 50 is rather low and it would be better to consider for a real-life scenario an IoU of 60 or 70. Then, however the model generally does not perform very well, which will hamper accurate down-stream single cell analysis.

Reviewer #2 (Remarks to the Author):

The paper presents a method to segment cell boundaries from weak annotations (background/foreground masks and cell centroids/point annotations) using an auxiliary network. There is a lot of work in this space already and unfortunately none of these works are cited or differentiated/compared against:

Khalid, N. et al. (2022). Point2Mask: A Weakly Supervised Approach for Cell Segmentation Using Point Annotation. In: Yang, G., Aviles-Rivero, A., Roberts, M., Schönlieb, CB. (eds) Medical Image Understanding and Analysis. MIUA 2022. Lecture Notes in Computer Science, vol 13413. Springer, Cham. https://doi.org/10.1007/978-3-031-12053-4_11

Cheng, Bowen, Omkar Parkhi, and Alexander Kirillov. "Pointly-supervised instance segmentation." In Proceedings of the IEEE/CVF Conference on Computer Vision and Pattern Recognition, pp. 2617-2626. 2022.

Laradji, Issam H., Negar Rostamzadeh, Pedro O. Pinheiro, David Vazquez, and Mark Schmidt. "Proposal-based instance segmentation with point supervision." In 2020 IEEE International Conference on Image Processing (ICIP), pp. 2126-2130. IEEE, 2020.

Khoreva, Anna, Rodrigo Benenson, Jan Hosang, Matthias Hein, and Bernt Schiele. "Simple does it: Weakly supervised instance and semantic segmentation." In Proceedings of the IEEE conference on computer vision and pattern recognition, pp. 876-885. 2017.

Saleh, Fatemehsadat, Mohammad Sadegh Aliakbarian, Mathieu Salzmann, Lars Petersson, Stephen Gould, and Jose M. Alvarez. "Built-in foreground/background prior for weakly-supervised semantic segmentation." In European conference on computer vision, pp. 413-432. Springer, Cham, 2016.

Unfortunately, the novelty is limited. Also there are important recent works in the space of whole-cell segmentation in fluorescence images that should be compared against, especially on their TissueNet dataset which is closer to real-world than the cell lines in Cellpose data:

Greenwald, Noah F., Geneva Miller, Erick Moen, Alex Kong, Adam Kagel, Thomas Dougherty, Christine Camacho Fullaway et al. "Whole-cell segmentation of tissue images with human-level performance using large-scale data annotation and deep learning." *Nature biotechnology* 40, no. 4 (2022): 555-565.

Also, Cellpose 2.0 does away with some of the pain with densely delineating the boundaries of cells in the image by using an efficient human-in-the-loop approach:

Stringer, Carsen, and Marius Pachitariu. "Cellpose 2.0: how to train your own model." *bioRxiv* (2022).

There are other auxiliary tasks such as self-supervised denoising (which when added to cell instance segmentation) requires minimum amount of data to achieve really good results:

Buchholz, Tim-Oliver, Mangal Prakash, Deborah Schmidt, Alexander Krull, and Florian Jug. "DenoSeg: joint denoising and segmentation." In *European Conference on Computer Vision*, pp. 324-337. Springer, Cham, 2020.

Also the reported metrics should take into account Aggregated Jaccard Index, mIOU, F1 as well.

Another important consideration one needs to keep in mind with respect to the background/foreground regions masks is in real-world datasets, background noise gradient is common. A single threshold might not be enough to separate the foreground/background pixels. In these cases, this region proposal network will not work.

Good luck going forward.

Response to reviewers' comments:

Below the reviews' critiques are in *italic*, and our responses are in normal text.

Reviewer #1 (Remarks to the Author):

The manuscript "Efficient End-to-end Learning for Cell Segmentation with Machine Generated Weak Annotations" by Shrestha et al, addresses the current need in cell image segmentation for deep learning algorithms to be trained on human annotator generated ground truths. The authors propose an end-to-end model using machine generated annotations and show that in some microscopy image data series segmentation results are equally or even more accurate as compared to previously published models, such as Cellpose. For some images AP is less than expected. The main strength of the approach is the author's notion to keep annotation times short. Further the authors propose to additionally predict cell boundaries using an auxiliary convolutional network which improves the result of weakly-supervised models.

I have two main points of criticism/suggestion:

1. The impact of the work would improve significantly if the value of the model would be tested on tissue images rather than solely cell cultures.

Response: We have now included benchmark results on the tissuenet dataset, which is currently the biggest single-cell segmentation dataset on tissue imaging (p9, lin 193). We show that our weakly-supervised model achieves ~95% of the accuracy of a fully-supervised model.

We originally avoided this dataset because of the low human-to-human consistency on this specific dataset. Thus a high model accuracy requires correct modeling of the annotation preferences of the specific annotator(s), which is more difficult to do with weakly supervised and unsupervised learning.

2. Figure 4 shows that at an IoU of 50 – which is rather low – the weakly supervised model plus auxnet (nearly) reaches AP levels of the supervised model. However, an IoU of 50 is rather low and it would be better to consider for a real-life scenario an IoU of 60 or 70. Then, however the model generally does not perform very well, which will hamper accurate down-stream single cell analysis.

Response: The figure below shows the relative performance of (weakly-supervised) LACSS model versus the fully-supervised models, based on the data shown in Figure. 4. It is evident that the relative performance of the weakly-supervised model remains relatively constant in a wide range of IoU value, and only exhibited a clear weakness against the fully-supervised model at much higher IoUs (e.g., > 0.8).

Reviewer #2 (Remarks to the Author):

1. The paper presents a method to segment cell boundaries from weak annotations (background/foreground masks and cell centroids/point annotations) using an auxiliary network. There is lot of work in this space already and unfortunately none of these works are cited or differentiated/compared against:

Khalid, N. et al. (2022). *Point2Mask: A Weakly Supervised Approach for Cell Segmentation Using Point Annotation*. In: Yang, G., Aviles-Rivero, A., Roberts, M., Schönlieb, CB. (eds) *Medical Image Understanding and Analysis. MIUA 2022. Lecture Notes in Computer Science*, vol 13413. Springer, Cham. https://doi.org/10.1007/978-3-031-12053-4_11

Cheng, Bowen, Omkar Parkhi, and Alexander Kirillov. "Pointly-supervised instance segmentation." In *Proceedings of the IEEE/CVF Conference on Computer Vision and Pattern Recognition*, pp. 2617-2626. 2022.

Laradji, Issam H., Negar Rostamzadeh, Pedro O. Pinheiro, David Vazquez, and Mark Schmidt. "Proposal-based instance segmentation with point supervision." In *2020 IEEE International Conference on Image Processing (ICIP)*, pp. 2126-2130. IEEE, 2020.

Khoreva, Anna, Rodrigo Benenson, Jan Hosang, Matthias Hein, and Bernt Schiele. "Simple does it: Weakly supervised instance and semantic segmentation." In *Proceedings of the IEEE conference on computer vision and pattern recognition*, pp. 876-885. 2017.

Saleh, Fatemehsadat, Mohammad Sadegh Aliakbarian, Mathieu Salzmann, Lars Petersson, Stephen Gould, and Jose M. Alvarez. "Built-in foreground/background prior for weakly-supervised semantic segmentation." In *European conference on computer vision*, pp. 413-432. Springer, Cham, 2016.

Response: We disagree with the reviewer's assessment.

Among the five papers referenced here, the #1, #2, and #4 (Khalid et. al, Cheng et. al, and Khoreva et. al) rely on bounding boxes as the primary instance annotation. Indeed, bounding-boxes-based weak supervision is a well-explored area, and it is well-known by the field that bounding-boxes are effective supervision and can achieve >90% of performance in comparison to full supervision. We discussed this and referenced #2 and #4 already in the original manuscript. What may have added to the confusion is probably the fact that #1 and #2 also used point supervision in addition to bounding-boxes to further improve the model accuracy by a small margin (a few percent points). It is important to note that this is fundamentally different from our approach, which tries to use point-supervision as the ONLY instance-level annotation.

Furthermore, #5 (Saleh et. al) employs image-level supervision to train a segmentation model. The method is very similar to ref 21-23 in our manuscript, the main difference being that the paper we cited works on biological images while Saleh et. al did not. Image-level supervision is an attractive approach, but so far it works mainly as a semantic segmentation method and to our knowledge there is no known strategy to adopt this method to segment dense cell populations, where many similar instances in contact with each other exist in the same image. Therefore we cannot make direct quantitative comparisons to them. Finally, #3 (Laradji et. al) is spiritually the most similar to our work, as it also attempts to use points as the only instance annotations to train a model. If anything, this paper illustrates exactly how difficult such a task is, as it achieves abysmal performances in instance segmentation: On COCO dataset, their model achieves an AP50 of 0.17, while the state-of-the-art is > 0.7 for supervised models. Such a performance makes it essentially useless for single-cell segmentation.

We have expanded the introduction section (line 47-65) to give a clearer methodology comparison of our method to the existing literature and highlight the exact novelty of our work.

2. Unfortunately, the novelty is limited. Also there are important recent works in the space of whole-cell segmentation in fluorescence images that should be compared against, especially on their TissueNet dataset which is closer to real-world than the cell lines in Cellpose data:

Greenwald, Noah F., Geneva Miller, Erick Moen, Alex Kong, Adam Kagel, Thomas Dougherty, Christine Camacho Fullaway et al. "Whole-cell segmentation of tissue images with human-level performance using large-scale data annotation and deep learning." Nature biotechnology 40, no. 4 (2022): 555-565.

Response: We have added benchmark data on TissueNet dataset ((p9, lin 193). We show that our weakly-supervised model achieved ~95% of the performance in comparison to the fully supervised model.

3. Also, Cellpose 2.0 does away with some of the pain with densely delineating the boundaries of cells in the image by using an efficient human-in-the-loop approach:

Stringer, Carsen, and Marius Pachitariu. "Cellpose 2.0: how to train your own model." bioRxiv (2022).

There are other auxiliary tasks such as self-supervised denoising (which when added to cell instance segmentation) requires minimum amount of data to achieve really good results:

Buchholz, Tim-Oliver, Mangal Prakash, Deborah Schmidt, Alexander Krull, and Florian Jug. "DenoSeg: joint denoising and segmentation." In European Conference on Computer Vision, pp. 324-337. Springer, Cham, 2020.

Response: Indeed, there are multiple efforts on various fronts to reduce the annotation burden, highlighting the significance of the problem at hand. However, we believe these other works did not diminish the merits of our study, as they focus on different stages of the computational pipelines than ours.

4. Also the reported metrics should take into account Aggregated Jaccard Index, mIOU, F1 as well.

Response: Aggregated Jaccard Index, mIOU, and F1 are widely used for semantic segmentation tasks, but there is no agreed upon definitions of these metrics for instance segmentation models. In microscopy literature, we are aware of two use cases of F1/mIoU etc: (1) for single-nuclei segmentation, the problem is typically treated as a semantic segmentation task, because there is little need to worry about instance contact and instance occlusion. In this case, the definition of F1/mIoU is straightforward. But the practice cannot be extended to whole-cell segmentation. (2) For true single-cell segmentation, some authors create a custom definition of F1 by using their own algorithm of instance matching (e.g. Greenwald, et al. Nature biotechnology 40:555), however the custom definition (and the underlying instance matching algorithm) has not widely been adopted outside the specific study. We prefer to stick with the current *de facto* standard metric for instance segmentation, average precision, and only use other metrics when needed (i.e, for comparison with existing literature data).

5. Another important consideration one needs to keep in mind with respect to the background/foreground regions masks is in real-world datasets, background noise gradient is common. A single threshold might not be enough to separate the foreground/background pixels. In these cases, this region proposal network will not work.

Response: In our opinion, image-level segmentation should be considered a solved problem. For images that cannot be segmented with level-setting (e.g., bright field images),

a model as simple as Random Forest (see ref. 35) performs well and requires very little supervision to train. Deep-learning based models can also be trained with very weak supervision (ref.18), or completely unsupervised (ref. 26). Even in the worst case, when the annotation needs to be generated manually, it still represents a significant time saving compared to the manual annotations at the instance level.

Reviewers' comments:

Reviewer #1 (Remarks to the Author):

The authors have adequately addressed my concerns and comments in the revised manuscript. It is appreciated that results on tissue images are now included.

Reviewer #4 (Remarks to the Author):

Here, the authors present an instance segmentation method for cell in microscopy images based on point annotations. Their results are impressive with some achieving performance approach fully-supervised counterparts. Although the authors have appropriately address the reviewer's previous comments, I have some additional comments which I believe may improve the paper. Please see below.

Major comments

Based on the concluding paragraph in the introduction, my impression is that the annotation generation method is what the authors would like to emphasize as what their study contributes to the existing literature. Upon examination of these methods, they seem relatively simple. Other than that, the instance segmentation method is not much different from others. Adding channel-wise attention to CNNs is commonplace. Additionally, the authors admit in their supplementary file that LPN is very similar to RPN, the only differences resulting from the fact that the data or labels are not available. There are also a few modifications that result in more memory efficiency. Finally, I don't believe that instance segmentation networks like MaskRCNN, FastRCNN, FasterRCNN, and panoptic networks utilize Huber loss, so in a sense, this is novel. The authors should perform an ablation experiment in which they demonstrate that the specific design choices they make result in better performance over some baseline.

There is a baseline the authors suggest is in the same spirit of the current study. Cheng, Bowen, Omkar Parkhi, and Alexander Kirillov. "Pointly-supervised instance segmentation." In Proceedings of the IEEE/CVF Conference on Computer Vision and Pattern Recognition, pp. 2617-2626. 2022. I agree with the previous reviewer that this method should be discussed, which the authors undoubtedly did. However, it is typical in deep learning application papers to compare the proposed method to previous methods. Therefore, despite the author's baseless assertion that poor performance on COCO implies poor performance on their cell segmentation dataset, they must provide a comparison with this method.

Typically, methods (example – lines 223-233), datasets, experimental design (example – lines 116-120), and discussion (example – lines 163-173) not included in results. Methods reside in the methods, datasets and experimental design reside in the methods, and discussion resides in the discussion. The example provided are not the only instances. Please heavily revise the results to include results only and the components I described into their appropriate sections.

The authors chose different hyperparameters for difference cell lines when benchmarking their method for the LIVECell dataset. If this is what previous studies did when applying MaskRCNN and CenterMask, then what the authors have done is fine. However, if the previous studies did not do this, then it would behoove the authors to reproduce those studies' experiments by optimizing the same hyperparameters. Otherwise, it is not a fair comparison. Please revise appropriately.

The discussion section is skimpy at best. The authors do not discuss their results in relation to previous work, shortcomings of their method, and possible future directions of their research. Perhaps these can be found in the results section? Please revise.

Minor comments

Can the authors provide the number of images in the LIVECell dataset?

What do the authors mean by random initialization of ResNet weights? What distribution was used?

Can the authors provide their results without auxnet in the main manuscript?

Comma splice on lines 38, 41, 56, 78

Run-on sentence on lines 44-46,

Y'all put references before periods but then after the comma on line 51.

Semi-colon inappropriate on line 62. Should be –

Colon inappropriate on lines 66, 267. Following sentence is an independent clause.

Can the authors clarify “novel model architecture that intelligently incorporates prior knowledge regarding the input data” on lines 62-63? This is a generic statement. Specify the prior knowledge.

“Lesser” on line 71 should be “fewer”.

Are you sure it's a 20x objective and 0.8 microns per pixel? Usually 20x results from 0.45-0.5 microns per pixel.

Below the reviewer's original comments are in **bold**. Our responses are in normal font.

Reviewer #4 (Remarks to the Author):

Major comments

- 1. Based on the concluding paragraph in the introduction, my impression is that the annotation generation method is what the authors would like to emphasize as what their study contributes to the existing literature. Upon examination of these methods, they seem relatively simple. Other than that, the instance segmentation method is not much different from others. Adding channel-wise attention to CNNs is commonplace. Additionally, the authors admit in their supplementary file that LPN is very similar to RPN, the only differences resulting from the fact that the data or labels are not available. There are also a few modifications that result in more memory efficiency. Finally, I don't believe that instance segmentation networks like MaskRCNN, FastRCNN, FasterRCNN, and panoptic networks utilize Huber loss, so in a sense, this is novel.**

We would argue that the paper made two major contributions.

The first contribution is demonstrating a new supervision structure for model training. We use LOIs plus image-level segmentation to train an instance segmentation model. We are not aware of any existing segmentation models that support such a supervision structure.

The second contribution is demonstrating a new technique of using a low-FOV CNN (auxnet) to regularize the model training. The supervision for LACSS is weak so that the model can easily converge to optima that perfectly satisfies the supervision constraints, yet produce poor quality segmentations. We showed in the paper that "auxnet" entices the model to converge to much better solutions. To our knowledge, this technique had not been used in previous segmentation models.

In addition, in terms of model structure, LACSS differs from other related models, such as MaskRCNN, by incorporating a novel positional encoding strategy. Positional encodings were used to break the translational invariability of the CNN, otherwise a CNN would never be able to establish the concept of "instances". MaskRCNN also needs to break the translational invariability, but uses a much simpler method, namely, cropping. LACSS, on the other hand, cannot crop because it is trained without bounding boxes.

We do not consider the use of Huber loss particularly noteworthy.

Unfortunately, the previous manuscript gave the reviewer the wrong impression that "annotation generation method" is the focus of the study. It is not. On the contrary, the point of the study is to better utilize annotations that we already know how to generate efficiently.

We have rewritten the introduction and expanded the discussion (line 272-287) to express the above points. These changes will hopefully help readers better understand the work submitted.

2. The authors should perform an ablation experiment in which they demonstrate that the specific design choices they make result in better performance over some baseline.

Ablation experiments are useful to evaluate novel changes to an existing method. There are three elements of the study that we consider novel (see answer to question 1):

- (1) A new supervision structure. Ablation study is not suitable for this type of change.
- (2) The use of auxnet. We already performed an ablation study regarding auxnet in the original manuscript. See Figure 4.
- (3) The use of positional encodings. While we cannot “ablate” positional encodings, we can compare different specific implementations. We show results comparing simple fixed linear encodings with learned encodings. See suppl. text and suppl. Fig. S5.

3. There is a baseline the authors suggest is in the same spirit of the current study. Cheng, Bowen, Omkar Parkhi, and Alexander Kirillov. "Pointly-supervised instance segmentation." In Proceedings of the IEEE/CVF Conference on Computer Vision and Pattern Recognition, pp. 2617-2626. 2022. I agree with the previous reviewer that this method should be discussed, which the authors undoubtedly did. However, it is typical in deep learning application papers to compare the proposed method to previous methods. Therefore, despite the author's baseless assertion that poor performance on COCO implies poor performance on their cell segmentation dataset, they must provide a comparison with this method.

The reviewer has confirmed that they were referring to the Wise-net method in “Laradji, et. al, Proposal-Based Instance Segmentation With Point Supervision. in ICIP, 2126–2130 (2020)”

We had attempted to establish new cell segmentation models using the published WISE-net code (<https://github.com/IssamLaradji/wisenet>). However, the code is no longer being maintained. Attempts of following the instructions in README to reproduce the original publication resulted in execution errors. We were able to reach the author by email, but so far the code issue has not been resolved yet.

Below we would like to provide alternative evidence that WISE-net is not suitable for the cell segmentation tasks we study in the paper.

A key idea of WISE-net is to use unsupervised super-pixel algorithms to generate a large amount of potential segmentation “proposals” (thus “proposal-based”), most of which are of course wrong. A deep learning model, however, can be trained to reject “bad” proposals, relying on the supervision of LOI labels. The remaining “good” proposals will be the model output. Therefore,

for the method to work, the algorithm needs to at least have a chance to generate the good proposals.

Below we show the recall rate of proposals generated on the LIVECell dataset using WISE-net's engine. We set the hyperparameters so that the engine produces ~20x more proposals than the ground truth instances. The x-axis is the IOU criteria for a proposal to be considered "good". Note that the model will only reject bad proposals and thus can only improve accuracy but not recall. It is easy to see that even if the model could achieve 100% accuracy, the overall performance would still be poor.

To us, this result is not surprising. Afterall, WISE-net is trying to solve a problem that is significantly harder than ours – instance segmentation using only LOI supervision. By the same argument, we also would not consider this method a "baseline" of ours, because it would be an unfair comparison.

- 4. Typically, methods (example – lines 223-233), datasets, experimental design (example – lines 116-120), and discussion (example – lines 163-173) not included in results. Methods reside in the methods, datasets and experimental design reside in the methods, and discussion resides in the discussion. The example provided are not the only instances. Please heavily revise the results to include results only and the components I described into their appropriate sections.**

We have restructured the text according to the reviewer's suggestion. (1) A new section, "dataset" was added to the method section. (2) Various discussions were removed from the results. Some are replicated in the discussion section (e.g., line 294-298).

- 5. The authors chose different hyperparameters for difference cell lines when benchmarking their method for the LIVECell dataset. If this is what previous studies did when applying MaskRCNN and CenterMask, then what the authors have done is fine.**

However, if the previous studies did not do this, then it would behoove the authors to reproduce those studies' experiments by optimizing the same hyperparameters. Otherwise, it is not a fair comparison. Please revise appropriately.

The LIVECell benchmarks are now based on the generalist LACSS model (see revised table 1). For LACSS, the generalist model underperforms the specialist models in AP50, albeit a slightly better in AP75 (see table below). The original LIVECell paper presented both a generalist model and specialist models, but the generalist model performed the best. We agree comparing generalist models with generalist models is the better choice.

	Generalist		Specialist	
	AP50	AP75	AP50	AP75
A172	68.0	28.8	68.3	27.3
BT474	68.3	36.2	69.1	34.1
BV2	87.7	57.9	86.5	57.1
Huh7	70.2	47.7	70.6	45.1
MCF7	71.2	26.5	72.2	25.5
SHSY5Y	43.2	5.9	47.2	6.3
SKOV3	81.6	48.0	85.6	49.7
SkBr3	93.1	80.7	93.1	80.0

- 6. The discussion section is skimpy at best. The authors do not discuss their results in relation to previous work, shortcomings of their method, and possible future directions of their research. Perhaps these can be found in the results section? Please revise.**

We have expanded the discussions (line 270-301) to highlight the differences of our approach with previous works and added discussions of limitations and future directions.

Minor comments

- 1. Can the authors provide the number of images in the LIVECell dataset?**

We added the numbers to the method section: 3253 training, 570 validation and 1564 testing. These images are quite large – can have up to 3000 cells per image.

- 2. What do the authors mean by random initialization of ResNet weights? What distribution was used?**

We used He initialization. The reference is added.

- 3. Can the authors provide their results without auxnet in the main manuscript?**

For LIVECell, results without “auxnet” had been shown in Figure 4. In addition, numerical values are now tabulated in suppl. Table S1. The revision now also showed comparisons for other datasets (see table 2 and table 3).

4. Comma splice on lines 38, 41, 56, 78

We corrected these as well as several other misused commas in front of the word “because”.

5. Run-on sentence on lines 44-46,

We shorten the sentence to:

This drawback further raises concerns over the scalability of the deep-learning method for three-dimensional (3D) microscopy, for which manual annotations are even more expensive to produce.

6. Y'all put references before periods but then after the comma on line 51.

The comma is removed.

7. Semi-colon inappropriate on line 62. Should be –

Changed to hyphen as recommended

8. Colon inappropriate on lines 66, 267. Following sentence is an independent clause.

Changed both to period.

9. Can the authors clarify “novel model architecture that intelligently incorporates prior knowledge regarding the input data” on lines 62-63? This is a generic statement. Specify the prior knowledge.

The sentence was changed to:

Another revenue of research is to incorporate specific prior knowledge into models built for specific microscopy modalities. For example, Hou et al were able to model H&E images by combining a sparse autoencoder to represent nuclei blobs and a light-weight CNN for background, and extracted useful feature representations based on unsupervised training

10. “Lesser” on line 71 should be “fewer”.

Changed to “fast annotation”.

11. Are you sure it's a 20x objective and 0.8 microns per pixel? Usually 20x results from 0.45-0.5 microns per pixel.

Yes. Our microscope is custom built and uses components that might be considered "atypical".

Reviewer #5 (Remarks to the Author):

- 1. Primarily, the authors put a heavy emphasis on the computer vision innovation (line 47 to 65) to layout the issue for the need of intensive costly annotation, and how recent methods trying to resolve it. While it is good, the author did not perform enough literature search on the problem for the cell segmentation. Many previous works have approached in similar direction, while might used different idea for cell segmentation, but those works were not cited. Some works used similar methods (e.g. using key point proposal in the network structure), those works were not cited neither. There is nothing wrong to have similar ideas if the pipeline that the author proposed advanced methods/knowledge in a different aspect. Citing relevant works in the field is important: 1) this will help the reader to understand the specific domain better and the major contribution of this work, 2) the reader can read through methods in the field of cell segmentation and find a proper method for their problem through this paper, 3) the previous works in the field should be properly acknowledged. The author primarily cites the general network papers. This is not a computer vision paper, and the author did not perform a substantial/innovated modification on the network structure. The paper will have a much better readability if the author can perform a through literature search and cite the relevant works in cell segmentation field. The following works should be cited and discussed (but the citation should not be limited to the list, more relevant works should be included in the citation and be discussed in the introduction/discussion):**

S. E. A. Raza et al. Micro-Net: A unified model for segmentation of various objects in microscopy images. Med. Image Anal.

A. Carpenter et al. CellProfiler: image analysis software for identifying and quantifying cell phenotypes. Genome Biol.

D. A. Van Valen et al. Deep learning automates the quantitative analysis of individual cells in live-cell imaging experiments. PLoS Comput. Biol.

W. Han et al. Cell segmentation for immunofluorescence multiplexed images using two-stage domain adaptation and weakly labeled data for pre-training, Scientific reports.

S. Graham et al. Hover-Net: Simultaneous segmentation and classification of nuclei in multi-tissue histology images. Med. Image Anal.

The introduction section has been expanded in an attempt to give a comprehensive review of the single-cell segmentation literature (line 38-75). Currently, the most widely used methods for

single-cell segmentation are based on image-to-image translation models (e.g. UNet). We previously gave relatively little discussion about these works because we had mainly focused on weakly-supervised literature, most of which are based on the object detection models. The revision now discussed both approaches.

- 2. The metrics in the paper are inconsistent. I understand that the author used AP for comparison purposes as it is a widely used metric for instance segmentation in computer vision application. However, for cell segmentation, AP does not tell us about how accurate the segmentation boundary aligns with the ground truth boundary. In some cases, such as quantifying the signals using the segmentation mask, the misalignment at the boundary might lead to a large error. It is important for the users to have results that uses relevant metrics for the problem. Benchmarking is a plus, but the purpose of using a relevant metric is more important to help the reader to figure out if the proposed method is suitable for their problem. Therefore, the author should also consider other relevant metrics (e.g. object dice or object Hausdorff). If other compared methods did not compute those metrics, the author can compute it for them for comparison purpose. Most previous methods have code available. For some methods that does not have code, the author can reproduce their method based on their understanding of the paper. The author should consider which metrics gives the best measurement for their problem/application.**

We now also report aggregated Jaccard index (AJI) for all LACSS models. AJI is similar to object Dice and is a slightly more popular metric in the cell/nuclei segmentation literature.

- 3. I am not sure if this is because the paper version that I have is a review version, the quality of the figures in this paper needs to be improved: the figures in this paper have low resolution. The resolution affects the readability (See Figure 2). Many words in the figure 2 are fuzzy, and I cannot figure out what are those words.**

We now submit high resolution figures in a separate ppt file to avoid rendering issues.

- 4. Importantly, I did not find the code link for the model proposed in this paper. Please provide the code link for purpose of review and reproducibility. The author should provide a github link (it can also be another repository) so that reviewers and readers can run the code to reproduce any results in the paper.**

The original manuscript had cited the code url (<https://github.com/jiyuuchc/lacss>) at line 320, near the end of the main text. See the “code availability” section.

Below are point by point recommendations:

1. In line 76-77: 'Nucleus detection from DAPI... are readily available'. Please provide citations to support the statement. Also, the citations helps the reader to find associated code that works well for DAPI segmentation.

We have rephrased the sentence and added references (line 109-111).

2. In line 84-95: Please cite the works that also used point proposal for cell segmentation in the literature. The authors should cite the previous works when they describe their network structure.

We would argue it is not helpful to make a direct reference to the point-supervision literature here because point supervision and LOI supervision are different from each other.

Point supervisions are *random* pixel samples of a segmentation mask and a much stronger form of supervision than LOI for segmentation. A simple case in point: Let's suppose the task is to segment dumbbells from images. For "point supervision", the labeled pixels will appear on the grip for some training images, and on the weights for others, collectively providing a relatively complete semantic definition of the dumbbell object. In contrast, LOIs will only label the grip, therefore do not clearly define the concept of "dumbbell". Furthermore, point supervision is exactly the type of annotations we try to avoid using, because it requires a human annotator.

While point supervision has been used in cell segmentation literature, to our knowledge, there has not been a published cell segmentation model that relies on LOI supervision. Outside the biological literature, we know only one paper (Laradji, I. H. et. al.) that uses LOI (but it is difficult to do an exhaustive search, because the terminology is not used consistently), which we had cited (ref 27). See also our answer to reviewer 4's comment 3 for related discussions regarding ref 27.

Additionally, point supervisions are used to train standard models designed for fully-supervised training. The only change is that one ignores all the pixels that are unlabeled when computing the loss function (see e.g. ref 26). In other words, there is no "network structure" that is specific to point supervisions. Thus it doesn't make sense to compare the network structure of LACSS with models using point supervisions, since it would simply be comparing LACSS with a standard model such as MaskRCNN (which we had already been doing).

We have now added more discussions regarding why the LOI supervision is different from the point supervision (line 83-84 and 307-313), as it seems to be a point of confusion.

3. In line 115-116: 'Therefore segmentation is relatively easy.' Please remove this statement as this is subjective and does not help the reader to understand anything.

We have removed the sentence.

4. In line 199-202: I recommend rephrasing the content. 'human annotator is about 20% lower than a machine model', I am not sure how this is computed, in what metric. Bear my ignorance, I can only imagine how one can compute inter-observer variability. However, how one can demonstrate the machine is performing better 20% than the annotator? A clearer explanation helps the understanding. Also, I recommend changing the wording for saying 'TissueNet Dataset is an ill-suited task'. This confuses the reader to understand the purpose of testing the model on this data. I believe testing on this data is a great demonstration to show your method performance on the TissueNet task.

We have rephrased the sentences. It now says:

Greenwald et. al¹⁶ had asked different human annotators to re-segment the TissueNet data, and compared the results against the "ground truth", and found that they typically achieve a score (F1~0.65) much lower than the fully-supervised CNN models (F1 > 0.8). This suggests that part of the model performance should be attributed to its ability to reproduce the annotator biases it was trained on. For weakly-supervised models, recapitulating annotator biases might be difficult because they don't have full access to all the annotations. We suspect this factor partly explains the performance gap between LACSS and other fully-supervised models.

We have also moved the paragraph to the discussion section per suggestion by reviewer 4.

5. In line 207-208: I am not convinced how a model perform better than the than human level performance. How did you exactly measure that? Did you have a gold standard annotator, and other annotators will validate against the gold standard? Or you meant the model have smaller inter-observer variability against different annotators? If the latter, I guess that you can only argue that your model have smaller inter-observer ability, which potentially can be more stable in application.

We have removed the sentence. Also see answer to the last question.

6. In line 234-236: please elaborate why AP50=0.84 can be considered as practical. Please provide citations which include studies to support such a statement. In cell profiling cases, AP50 = 0.84 can be an unacceptable result for quantifying cell signals between nuclei and membrane boundaries.

We have removed the sentence.

7. In line 245-247: please provide citation for 'most of the existing works.' Also, I recommend removing 'which we believe may not be the most productive venue.' This is subjective and the authors did not provide a reason to support their statement.

We have rephrased the sentence. It now says:

One way to solve the problem is to use models that can be trained with weak supervision, although one needs to strike a balance between the model accuracy and the amount of annotation needed.

8. In line 274: please cite FIJI here.

Cited.

9. In line 285: please provide the name of the random algorithm which you used for initializing the weights and biases.

We use He initialization. Citation was added.

10. In line 296: The equation is incomplete.

Can the reviewer clarify? It is not obvious to us what is missing. If it is a rendering issue, the intended equation is this:

$$AP = \frac{1}{c} \sum_{k=1}^n P(k)T(k)$$

11. In line 307: please using proper mathematical format to define FNR. The numerator should not be a phrase with a special character.

We have removed the FNR metrics from the paper in favor of AJI, and in order to be consistent among different datasets.

12. In line 313-315: Where are the statistics? How the description here supports reproducibility? Taking a sample for figures should not be included here. Did you use seed for initialization so that the results can be reproduced? Or, you might just remove this subsection as it does not include any information that is relevant to this subsection title.

We believe the section is mandatory. We have rephrased the section. It now says:

The study did not employ statistical analysis when comparing LACSS models with published baselines. All reported model benchmarks are based on best models obtained from multiple reruns of the training pipelines. Model checkpoints are provided (see Data Availability) to allow re-evaluation.

Tensorflow has support for deterministic computing mode but the speed is slower. We did not enable the deterministic mode in our model training.

13. In line 360-369 in Figure 6: I recommend adding one more row of predicted segmentation to a and b. Only provide ground truth figure is pointless. A comparison of ground truth to the segmentation results are necessary. Also, those figure sets are too small to see the details. The figure should be much larger for readability. If you are short of space, c and d can be moved to a different row.

The figure is meant to show comparison between the ground-truth (color maps) and model predictions (white outlines) (see figure caption). The presentation is the same as that in Figure 5. There might be a problem with the image resolution making the white outline difficult to see for the reviewer. In the revision all figures are now in a separate ppt file to avoid rendering issues.

14. In Table1 - Table 3 in page 22-24: the font in the two tables is inconsistently formatted, which some are left aligned, and some are right aligned.

The alignment is now consistent.

15. In Figure 7: captions in each column are not in the same relative position. Some are at the left; some are in the middle. Please make sure the captions are properly formatted. In b, in the last column, which one is the manual annotation, and which one is the prediction? You need to clarify that in the figure captions. Are the lines manual annotation? The lines are very low contrast to see.

We have adjusted the alignment.

We have updated the figure caption for 7b. Here the color maps are the ground truth, and the white contours are the model predictions.

REVIEWERS' COMMENTS:

Reviewer #4 (Remarks to the Author):

All of my concerns have essentially been addressed by the authors.

Reviewer #5 (Remarks to the Author):

The authors did a great job for resolving my comments and concerns regarding the revision. However, there are small format issues that may need to be resolved before publication. Please see the details below:

line 364: the equation is still incomplete. I am not sure if it is the rendering issue. I still see an incomplete/broken equation in the merged paper file while the equation in the revision file is fine. It seems we had similar issues before when I was reviewing the previous versions. I guess that the author/journal might want to carefully check the final version before publication to avoid the unnecessary surprises.

For the format, I prefer the Table 2 style, which looks clean. I suggest the author to convert Table 1 and 3 to the same style as Table 2. Also, it would be better if the author just uses two decimals in Table 1 and 3. This will be consistent to Table 2. Otherwise, the format does not align. Some tables have two decimal numbers while some have three. If the authors have specific reasons for that, could you clarify?

In page 20 (the reference page), from citation 10 to the end, the format is off. The reference list does not have a large space for reference 1 to 9. However, from 10 to the end, there is a large space after the reference number. Please make sure the format is consistent for all the references and comply with journal's requirement.

Below the reviewer's original comments are in **bold**. Our responses are in normal font.

Reviewer #5 (Remarks to the Author):

line 364: the equation is still incomplete. I am not sure if it is the rendering issue. I still see an incomplete/broken equation in the merged paper file while the equation in the revision file is fine. It seems we had similar issues before when I was reviewing the previous versions. I guess that the author/journal might want to carefully check the final version before publication to avoid the unnecessary surprises.

We have changed the font used for the equation rendering and doubled checked the final rendering.

For the format, I prefer the Table 2 style, which looks clean. I suggest the author to convert Table 1 and 3 to the same style as Table 2. Also, it would be better if the author just uses two decimals in Table 1 and 3. This will be consistent to Table 2. Otherwise, the format does not align. Some tables have two decimal numbers while some have three. If the authors have specific reasons for that, could you clarify?

We have changed all Tables to have the consistent style as suggested. The decimal number of Table 1 was changed to two.

In page 20 (the reference page), from citation 10 to the end, the format is off. The reference list does not have a large space for reference 1 to 9. However, from 10 to the end, there is a large space after the reference number. Please make sure the format is consistent for all the references and comply with journal's requirement.

We have reformatted the reference.